# Measuring the Microscopic Structures of Human Dental Enamel Can Predict Caries Experience

**DOI:** 10.3390/jpm10010005

**Published:** 2020-02-02

**Authors:** Ariana M. Kelly, Anna Kallistova, Erika C. Küchler, Helena F. Romanos, Andrea Lips, Marcelo C. Costa, Adriana Modesto, Alexandre R. Vieira

**Affiliations:** 1Department of Oral Biology, School of Dental Medicine, University of Pittsburgh, Pittsburgh, PA 15213, USA; amk211@pitt.edu (A.M.K.); erikacalvano@gmail.com (E.C.K.); 2Institute of Geochemistry, Mineralogy and Mineral Resources, Faculty of Science, Charles University, Albertov 6, Prague 2, Czech; kallistova.anna@gmail.com; 3Institute of Geology of the CAS, v.v.i., Rozvojová 269, Prague 6, Czech; 4Department of Pediatric Dentistry and Orthodontics, Federal University of Rio de Janeiro, Rio de Janeiro, RJ 21941-901, Brazil; lenafrr@hotmail.com (H.F.R.); pttpo2009@yahoo.com.br (M.C.C.); 5Clinical Research Unit, Fluminense Federal University, Niteról 24020, Brazil; alipsuff@hotmail.com; 6Department of Pediatric Dentistry, School of Dental Medicine, University of Pittsburgh, Pittsburgh, PA 15213, USA; ams208@pitt.edu

**Keywords:** dental caries, dental enamel, amelogenesis, genetics, polymorphism

## Abstract

Objectives: The hierarchical structure of enamel gives insight on the properties of enamel and can influence its strength and ultimately caries experience. Currently, past caries experience is quantified using the decayed, missing, filled teeth/decayed, missing, filled surface (DMFT/DMFS for permanent teeth; dmft/dmfs for primary teeth), or international caries detection and assessment system (ICDAS) scores. By analyzing the structure of enamel, a new measurement can be utilized clinically to predict susceptibility to future caries experience based on a patient’s individual’s biomarkers. The purpose of this study was to test the hypothesis that number of prisms by square millimeter in enamel and average gap distance between prisms and interprismatic areas, influence caries experience through genetic variation of the genes involved in enamel formation. Materials and Methods: Scanning electron microscopy (SEM) images of enamel from primary teeth were used to measure (i) number of prisms by square millimeter and interprismatic spaces, (ii) prism density, and (iii) gap distances between prisms in the enamel samples. The measurements were tested to explore a genetic association with variants of selected genes and correlations with caries experience based on the individual’s DMFT+ dmft score and enamel microhardness at baseline, after an artificial lesion was created and after the artificial lesion was treated with fluoride. Results: Associations were found between variants of genes including ameloblastin, amelogenin, enamelin, tuftelin, tuftelin interactive protein 11, beta defensin 1, matrix metallopeptidase 20 and enamel structure variables measured (number of prisms by square millimeter in enamel and average gap distance between prisms and interprismatic areas). Significant correlations were found between caries experience and microhardness and enamel structure. Negative correlations were found between number of prisms by square millimeter and high caries experience (r value= −0.71), gap distance between prisms and the enamel microhardness after an artificial lesion was created (r value= −0.70), and gap distance between prisms and the enamel microhardness after an artificial lesion was created and then treated with fluoride (r value= −0.81). There was a positive correlation between number of prisms by square millimeter and prism density of the enamel (r value = 0.82). Conclusions: Our data support that genetic variation may impact enamel formation, and therefore influence susceptibility to dental caries and future caries experience. Clinical Relevance: The evaluation of enamel structure that may impact caries experience allows for hypothesizing that the identification of individuals at higher risk for dental caries and implementation of personalized preventative treatments may one day become a reality.

## 1. Introduction

Previous studies have shown a genetic association between dental caries and polymorphisms in genes associated with the mineralization of bone, formation of enamel, microbial colonization and the degradation of amelogenin, which is an essential step in enamel formation [1,2,3,4,5]. Other studies have shown that the formation of dental enamel is influenced by genetic variation and can impact the prevalence of dental caries [6]. Further, it has also been shown that mouse dental enamel with less amelogenin is “weaker” than enamel with normal levels of amelogenin [7]. The limitation of many association studies with humans has been the oversimplified definition of the phenotype. The majority of studies compare individuals with no caries lesions with individuals with any number of caries lesions. We have attempted to create more discreet phenotypes (separating individuals with just a few caries lesions from individuals with many more lesions) or to use caries experience scores as a continuous variable (reviewed in [5,8]) and have proposed to use different phenotypical characterizations, such as the efficiency of creating artificial sub-clinical caries lesions in the laboratory having a corresponding DNA sample from the individual who donated the enamel specimen [1,2]. We think the structure of dental enamel can also explain the unique properties of enamel as well as the difference in prevalence of dental caries from person to person. 

From the analysis of micrographs of human dental enamel, a seven-level hierarchical structure of enamel was generated [8]. Starting at the nanoscale, the hierarchy begins with hydroxyapatite crystals, which build upon each other to form nanofibrils. These nanofibrils then aggregate to form thick fibrils and the thick fibrils form thicker fibers, termed crystal fibers. The crystal fibers then cluster together in two different orientations to form the prism/interprism continua. The prisms and interprismatic areas assemble into bands, which are oriented differently across the entire layer of enamel and finally the enamel layer is formed [9]. 

Prisms, formed from hydroxyapatite crystals, are the structural blocks of enamel. The structure of enamel is discontinuous where the prisms and interprismatic spaces meet, leaving a gap [9]. We hypothesize that if a person has more prisms and less interprismatic spaces, their enamel is denser and, therefore, “stronger” against acidic conditions and we hypothesize less susceptible to dental caries. Additionally, we further hypothesize that the size of the gap (prism sheath) between the prisms can be revealing of enamel strength. If a person has larger gaps, there is less enamel and therefore, the enamel is potentially “weaker against acidic challenges”, and we hypothesize more vulnerable to decay, although gaps may contribute to mechanical strength and toughness. Therefore, the structure of enamel may help us predict the strength of the enamel, proneness to dental caries, and future caries experience. Until now, caries experience was most commonly evaluated in genetic studies based on the count of the number of decayed, missing due to caries, filled teeth/decayed, missing due to caries, filled surface (DMFT/DMFS score for permanent teeth; dmft/dmfs score for primary teeth). It is not known the predictive value of measuring dental enamel structural features and how they may correlate with genomics. Therefore, measuring the number of prisms by square millimeter and interprismatic spaces in enamel, the prism density of the enamel and the size of the gaps between prisms and their association with caries experience has never before been explored. We used SEM images of primary molars to generate measurements related to the structure of enamel to test how genetic variation (if individuals were more likely to carry specific alleles) can impact individual susceptibility to dental caries in the permanent dentition. The design is inspired in the evidence that caries experience in the primary dentition can predict to a certain degree caries experience in the permanent dentition [10], as we still do not have a predictor for caries experience that does not require having had the disease in the past. Here, we tested if microscopic definitions of the enamel surface associated with genetic variation in enamel forming genes, with the idea that genomic data could serve as surrogate of the characteristics of the enamel surface and how it is susceptible to demineralization. 

## 2. Materials and Methods

To allow for phenotypical definitions that go beyond caries experience scores, we included in these analysis definitions previously obtained [1] from enamel microhardness measurements at baseline, after creation of an artificial sub-clinical caries lesions in the laboratory, and after the application of a fluoridated solution made from toothpaste. Furthermore, we used scanning electron microscopy to obtain new measurements of the enamel structure that were used in all analyses as described below. 

### 2.1. Enamel Specimens

Enamel samples from 108 exfoliated primary teeth (74 molars, 27 incisors, and seven canines) and genomic DNA from the 108 participants (mean age 10 years) who donated their sound teeth to participate in clinical dental research were available for this experiment, which is summarized in Figure 1.

All donated teeth were included in this study. Enamel samples did not include developmental defects, as that could be a confounder for all a posteriori analyses. The participant population consists of all Brazilian subjects from the same region, in an attempt to reduce the influence of environmental factors. We did not have data on fluoride exposure, professional fluoride treatments in sound surfaces, dietary habits, or toothbrushing frequency. Saliva samples were collected and written informed consent was obtained from all individual participants’ parents. This part of the study was approved by the University of Pittsburgh Institutional Review Board (IRB# 11070236) and by the Federal University of Rio de Janeiro (#333.167). Samples were collected by three examiners (E.C.K., A.L., and H.F.R.) and they were calibrated for assessing caries status by an experienced specialist (M.C.C.) [1]. The intra-examiner agreement was assessed by a second clinical examination in 10% of the sample after 2 weeks, with a j of 1.0 obtained. The Cohen’s kappa value for agreement between examiners was 0.91. The DMFT value was calculated for each subject [11], for both primary (dmft) and permanent (DMFT) dentitions. Teeth lost to trauma, or exfoliated primary teeth, were not included in the final DMFT/dmft scores. When records indicated that teeth were extracted for orthodontic reasons, or treatments were performed in sound teeth, these teeth were not included in the final DMFT/dmft scores [1]. Twenty-eight of the 74 molars that had a sound occlusal surface were selected and used for further microscopic analysis since they were considered to be sectioned at the same orientation. Characteristics of the population studied are presented in Table 1.

### 2.2. DNA Samples and Genotyping

Genomic DNA for molecular analysis was extracted from buccal cells [12]. Sixty single-nucleotide polymorphisms (SNPs) were selected and are listed in Table 2. These SNPs were chosen based on their locations relative to the genes and results of previous association studies with dental caries [1,2,3,4,13,14,15,16]. Polymerase chain reactions with TaqMan SNP Genotyping Assays from Applied Biosystems (Valencia, CA, USA), with a total volume of 3 µl per reaction and 3.0 ng of DNA per reaction, were used for genotyping all selected markers in a Tetrad PTC225 thermocycler from MJ Research (Waltham, MA, USA). Genotype detection and analysis were performed using the ABI 7900HT with ABI SDS software (Applied Biosystems, Valencia, CA, USA) [1].

### 2.3. Correlation Tests with Caries Experience and Enamel Microhardness as Phenotypes

The subjects were classified as having “low caries experience” (below the mean DMFT+dmft score of the 28 subjects) or “high caries experience” (above the mean DMFT+dmft score of the 28 subjects). A Pearson correlation test was used to evaluate the relationship between each phenotype obtained from the images and caries experience based on the DMFT+dmft score of each individual. Enamel microhardness data [1] were obtained for 17 of the 28 samples. Tooth samples were stored in 2% formaldehyde at room temperature for a few weeks until initial laboratory manipulation. At the start of sample preparations, the enamel surface was polished and blocks were cross-sectioned at one millimeter from the edge of each tooth sample to obtain 57 blocks of 3 × 4 × 3 millimeters from the buccal (N = 10), occlusal (N = 12), lingual (N = 5), mesial (N = 19) and distal (N = 15) surfaces. Blocks were submitted to baseline microhardness analysis at their surfaces using a microhardness tester (IndentaMet 1100 Series, Buehler Ltd., Lake Bluff, IL, USA) with a knoop diamond under a load of 25 grams for 10 seconds. Three indentations spaced 100 µm away from each other were made. Artificial carious lesions were created by immersing each enamel block in 24 mL of demineralizing solution (1.3 mmol/L Ca, 0.78 mmol/L P, 0.05 mol/L acetate buffer, 0.03 µgF/mL, pH 5.0) at 37 °C for 16 h. This method [2] produces a subsurface enamel demineralization without surface erosion. Surface microhardness was measured again by another three indentations created below the initial ones. Carious lesions were exposed to a fluoride solution made from toothpaste containing sodium fluoride (0.15% *w*/*v* fluoride ion; Aquafresh Extreme Clean, GlaxoSmithKline, Brentford, Middlesex, UK) to attempt to bring microhardness values back to baseline levels. This evaluation allows for determining if fluoride reuptake commensurate with baseline levels of microhardness and to help interpret how individual variation may impact findings. The toothpaste contains calcium and phosphate, and remineralization can be expected. Surface microhardness was measured one more time and other three indentations created right underneath the previous ones were obtained. After remineralization of the enamel block surfaces, a pH-cycling protocol was implemented to test the dynamic effect of fluoride on a high caries environmental challenge. The cycle alternated between a demineralizing solution (2.0 mmol/L Ca and P, 0.075 mol/L acetate buffer, 0.03 µgF/mL. pH 4.7; 0.75 mL/mm^2^) and a remineralizing solution (1.5 mmol/L Ca, 0.9 mm0l/L P, 0.15 mol/L KCl, 0.02 mol/L cacodylate buffer, 0.04 µg F/mL, pH 7.0; 0.25 mL/mm^2^) for 14 days. At 8 AM, all specimens were immersed in the remineralizing solution; at 12 PM, specimens were washed with deionized water and immersed in the demineralizing solution; at 2 PM, specimens were washed and immersed in the same remineralizing solution used at 8 AM; at 4 PM, specimens were washed and immersed in a new remineralizing solution, in which they were kept until 8 AM the next day, when the remineralizing solution was replaced again as a new cycle started. Surface microhardness was measured one last time and the measurement for the other three indentations created right underneath the ones previously obtained.

The baseline microhardness and rates of change of microhardness scores after artificial caries creation, fluoride application, and pH-cycling were calculated. Each measurement had three replicates and the mean of the three values was calculated and used in all analyses [1]. Phenotypes were defined based on dichotomous groups (baseline values or rate changes above or below the average of the group). These phenotypes were used to run the same correlation test and analyze the relationship between the measured phenotypes and the enamel microhardness at baseline, after the creation of a caries lesion, and after exposure to a fluoridated solution. Finally, a Pearson correlation test was used to evaluate the relationship between all three phenotypes.

### 2.4. Measurements of the Enamel Structures

A total of 445 scanning electron microscopy (SEM) images of enamel were obtained of 108 enamel specimens from 108 participants, one specimen per participant. Seventy-four of the 108 specimens were from molars, and 28 of the 74 molar specimens had sound occlusal surfaces. Multiple SEM images from each of the 28 molars were used for further analysis. The specimens were initially flattened with silicon carbide paper grit 320 and then further flattened by the use of 600 and 1200 grades of Al_2_O_3_ paper. The direction of a tooth for indentation was selected so that a polished surface parallel to the external surface was exposed for testing, which means the loading direction was parallel to the long axis of the enamel rods. Sections were done mesio-distally (Figure 2). Specimens were then polished to 1 μm with felt paper wet by 1 μm diamond spray. The polished samples were etched by 3% HCl (10 s), and then carbon sputtered. All specimens were mounted in the same orientation to allow that approximately the same areas were being captured. The images were collected in the secondary electron (SE) mode, in high vacuum at 20kV and at working distance of 8 to 10 mm using a Tescan Vega3 XMU at the Institute of Geology of the CAS, v.v.i. The smaller-scale images (greater than 100 nm) were excluded. Using *ImageJ* Image Processing and Analysis software [17], all of the included images were scaled to millimeters (mm) using the set scale function. Next, the areas of the included images were measured by selecting the image field and using the measure function on *ImageJ*. The number of prisms by square millimeter and interprismatic spaces in the image were measured by using the analyze particles function of *ImageJ.* The initial image (Figure 3a) was first made binary (Figure 3b) to normalize the color of the image and increase the accuracy of the measurement. After using the analyze particles tool on *ImageJ*, the number of prisms by square millimeter and interprismatic spaces in each image was recorded.

The prism density of the enamel was calculated by dividing the number of prisms by square millimeter by the area of enamel in the image. This is not the same of a measure of mineral density, which would require an examination of the inter-rod spaces, which are also mineralized. Additionally, the distance of the gaps between prisms (the interprismatic enamel) was measured. Using the draw line feature on *ImageJ*, a line was drawn from one end of the gap to the other end (Figure 4) and the measure function was used to record the length of the line. This was repeated ten times and the ten distances were averaged.

Once all of the measurements were recorded, the data were split into categories based on tooth (first and second upper molar and first and second lower molar) before analysis. Another category including all of data together was created.

### 2.5. Phenotype Definitions and Statistical Analysis

The subjects were classified as having a “low number of prisms by square millimeter” (below the mean number of prisms of the 28 subjects) or having a “high number of prisms by square millimeter” (above the mean number of prisms by square millimeter of the 28 subjects) (Figure 5). The number of these prisms was classified as phenotype 1. Similarly, the subjects were classified as having “less dense” enamel (below the mean prism density of the 28 subjects) or having “denser” enamel (above the mean prism density of the 28 subjects) (Figure 6a,b). The prism density was classified as phenotype 2 (calculated by dividing the number of prisms by square millimeter by the area of enamel in the image). Finally, the subjects were classified as having “small gaps” (below the mean gap distance of the 28 subjects) or having “large gaps” (above the mean gap distance of the 28 subjects) (Figure 7). The gap size was classified as phenotype 3. The samples were divided into categories based on the size of the SEM image and the tooth. This was used to explore potential differences depending on the tooth or the magnification used. The categories include: all data, first upper molar (1ST MU), second upper molar (2ND MU), first lower molar (1ST ML), and second lower molar (2nd ML). The genotyping data associated with the samples were used to test the allele frequencies between the phenotypes using the PLINK software package [18] with an alpha of 0.05. Standard case/control association analysis using Fisher’s exact test, as well as a Hardy–Weinberg equilibrium quality control test was used to analyze the data. Finally, we also performed linear regression as implemented in PLINK to test if caries experiences scores and enamel microhardness values associated with genotyping distributions and allele frequencies.

## 3. Results

Genotypes of all markers were in Hardy–Weinberg equilibrium. Complete results from the association and correlation analyses are listed in the appendix. 

### 3.1. Association Analyses

Associations between number of prisms by square millimeter, prism density, and gap distance and markers in multiple genes previously associated with dental caries and enamel formation (Table 3) were found, including *TUFT1*, *AMBN*, *KLK4*, *MMP20*, *ENAM*, and *DEFB1*. Two outliers that were included in the initial association tests were removed and the resulting data is presented in Table 3. Results from linear regression analyses were not significant (data not shown).

### 3.2. Correlation Analyses

Additionally, Pearson correlation tests between caries experience and the three phenotypes (the number of prisms by square millimeter was classified as phenotype 1, the prism density was classified as phenotype 2, and the gap size was classified as phenotype 3) showed significant correlations. There was a negative correlation (*r* = −0.58) between high caries experience and number of prisms by square millimeter in the 2nd MU category (Table 4). The correlation tests between enamel microhardness at the three stages (baseline, after creation of a caries lesion, and after fluoride treatment), and the three phenotypes showed multiple significant correlations. There was a negative correlation between baseline enamel microhardness and prism density of nanofibrils in all categories. Additionally, negative correlations were seen between enamel microhardness after lesion creation and gap size and enamel microhardness after fluoride treatment and gap size in 2nd MU categories (Table 4). There was a negative correlation between phenotype 2 and phenotype 3 in the all data set, a negative correlation between phenotypes 2 and 3 in the 1st MU data set, and finally a negative correlation between phenotype 2 and phenotype 3 in the 2nd MU data set.

The categories include all data, first upper molar (1ST MU), second upper molar (2ND MU), first lower molar (1ST ML), and second lower molar (2nd ML). The number of prisms by square millimeter was classified as phenotype 1 (P1). The prism density was classified as phenotype 2 (P2). The gap size was classified as phenotype 3 (P3).

## 4. Discussion

As previously stated, most studies thus far have quantified caries experience based on the DMFT score of the individual. Our previously published targeted genetic association tests with the DMFT+dmft scores (reviewed in [5]) showed that caries experience is associated to genetic variation in *AMBN*, *AMELX*, *ENAM*, *TFIP11*, *TUFT1*, *AQP5*, *KLK4*, *ALOX15*, *DEFB1*, and *ESRRB*. Interestingly, association studies using a hypothesis free approach (i.e., genome-wide searches) do not consistently show associations with the genes described above (reviewed in [5]) and we attribute this to the fact that the caries experience phenotype in these studies is defined as present or absent (caries free versus caries affected). Therefore, individuals with one lesion are combined with individuals with several lesions and these differences in disease presentation are relevant but disregarded and results from these studies have not been replicated. Using a new measurement to evaluate dental caries susceptibility, we were able to show genetic associations between the genes listed above and the number of prisms by square millimeter and prism density as well as the average gap distance between the prisms. When analyzing the all data set for phenotype 2 (prism density), two outliers were included in the analysis. As these two data points shifted the mean upward, the outliers were removed and further analysis was done to test the effects of removing the data points (Table 3). After removal, the heterogeneity decreased in the resulting population and this implies a greater genetic influence of these two outliers [5]. One caveat in our analyses is that occlusal surfaces have many angles and there is no way to standardize for the inclination of the enamel when the samples were polished, and to minimize this, we tried to always use the flattened area of the available surface. The remarkable aspect of the work however is that associations found here can be direct tested as possible predictors of dental caries experience. Most of the literature so far exploring the genetics of dental caries, in special genome-wide association studies (GWAS), by genotyping common genetic variants in the population and have used a very crude measurement for the phenotype (having had at least one dental caries lesion versus never having had a lesion), which tells nothing about the pathogenesis of dental caries in the individual. The most recent example is a meta-analysis of studies that include dental caries assessments ranging from self-reported questionnaire to standard clinical examinations for more than 13,000 children 2 to 18 years of age. The reduction of the phenotype to having at least one lesion versus having no lesions from scores obtained in primary and permanent dentitions led, not surprisingly, to meaningless results [19]. Here, we are reporting how differences observed in the enamel surface measured through scanning electron microscopy without any known bias on how these measurements were generated associate with genetic variants in genes previously associated with dental caries from hypothesis driven work. These phenotypes were measured in the occlusal surface of primary molars, which is typically the surface first affected in posterior teeth. The underlying assumption was that this would represent individual overall risk for caries and potentially allow the identification of associations that can translate in a tool for determination of risks early in life. 

As the power of the genetic effects of a test is dependent on the minor allele frequency (MAF) [20], we found that those individuals with the minor allele for a particular SNP and a statistically significant *p*-value had lower caries experience with more prisms, prism density, and smaller gaps and higher caries experience with less prisms, lower prism density, and larger gaps. One example of this was shown in the all data set for phenotype 3 and rs2619112, which is linked to *ALOX15*. Ten of the twenty-eight individuals with rs2619112 had the minor allele (G) and also had high caries experience with larger gap size or low caries experience with smaller gap size. As shown in Table 5, 94.25% of individuals with the minor allele and statistically significant associations to our phenotypes match our hypothesis regarding number of prisms by square millimeter, prism density, and gap distance and caries experience. Therefore, our data support the hypothesis that the structure of enamel, namely number of prisms by square millimeter density, and gap distance, play a role in the prevalence and the development of dental caries. 

Our results from correlation tests show an association between caries experience and the phenotypes. In the second upper molar data set, the greater number of prisms by square millimeter was negatively correlated (r = −0.72) with high caries experience, meaning that as the number of prisms by square millimeter increases, high caries experience decreases and vice-versa. In addition to correlations with caries experience, the phenotypes showed significant correlations with the enamel microhardness data at baseline, after the creation of a caries lesion, and after fluoride treatment. A similar negative correlation was shown in the second upper molar data set between average gap distance and enamel microhardness after caries lesion (r = −0.70) and after fluoride treatment (r = −0.81). These correlations show that as the average gap distance increases, the enamel microhardness after creation of a lesion and after fluoride treatment decreases and vice versa. Additionally, two positive correlations were found between greater number of prisms by square millimeter and enamel microhardness after caries lesion (r = 0.61) and after fluoride treatment (r = 0.61), showing that a greater number of prisms by square millimeter is correlated with higher values of enamel microhardness at these two stages. The correlation tests between our three phenotypes showed promising results. The negative correlation between phenotype 2 and 3 in the all data category (r = −0.46), the first upper molar category (r = −0.45), and the second upper molar category (r = −0.46), show that as prism density increases the gap distance decreases. The correlation data is important in showing a relationship between enamel structure and caries experience and the enamel microhardness of enamel, which can play a role in dental caries. Such a relationship is expected based on our hypothesis. In addition, we originally noticed [1] that the enamel microhardness values were lower than the ones typically reported for permanent teeth. This agrees with the general sense that primary teeth are more porous and therefore more susceptible to demineralization in comparison to permanent teeth due to greater mean prism-junction density and mean volume fraction of interprismatic enamel, as measured by planimetry of scanning electron micrographs [21,22,23].

Note that there are some limitations with our study. First, the teeth studied here have an unknown history in terms of fluoride or other exposures. They are also form a single cohort from an only one geographic area. Further, there is an inherited challenge in preparing specimens for these experiments, including polishing and sectioning them. There was a lack of enamel microhardness data for all of the samples. Only 18 of the 28 samples had enamel microhardness data associated with them. Further, knoop hardness is a relatively crude scale and other approaches such as nanoindentation testing may provide additional insight. Second, there was a lack of dmft (primary teeth) scores for the samples. As the samples were all primary teeth, ideally only the dmft values should have been used, but the combined DMFT+dmft scores (primary and permanent dentitions) were the only data available for the samples. Therefore, when classifying an individual as having low caries experience or high caries experience, the DMFT+dmft scores were used. Also, the quality of some of the images made measuring the enamel structures difficult and they had to be excluded, further limiting the sample size. Finally, the fixation of specimens in formaldehyde, which was done in a very low concentration, may alter infrared and Raman spectroscopy [24], and potentially alter the enamel surface [25]. These changes may impact the ability to detect matrix proteins and cause loss of water and it is difficult to imagine how this may have impacted the measurements we performed in this study. For hazardous reasons, we are not permitted in our laboratory to keep tooth specimens in natura and are required to fix them somehow.

In conclusion, our data support that genetic variation may impact enamel formation, and therefore influence susceptibility to dental caries and future caries experience. Our previous study has shown that results based on evaluations of enamel microhardness are distinct between permanent and primary teeth [1], and we hope to evaluate permanent teeth in the future. Additionally, enamel microhardness did not correlate with caries experience in our studies, but we found several correlations with enamel variation in the number of prisms by square millimeter, prism density, and size of gaps [1]. Our new way of analyzing the prevalence of dental caries goes beyond the traditional DMFT score and may provide a new tool for determining risk that should be further explored independently by others. By measuring the number of prisms by square millimeter, prism density, and average gap distance in enamel on a microscopic level, the prevalence of dental caries can be predicted earlier and therefore it may allow for a new strategy of preventing decayed, missing or filled teeth. For the first time a genetic component is brought to the measurement of caries prevalence. A genetic association was seen between our measurements and genes variants associated with dental caries. Although the results of the present study were found for primary teeth, these results can be extrapolated to permanent teeth, as an individual’s genetic make-up does not change, and there is a positive association between caries experience in primary teeth and caries experience in newly erupted permanent teeth [26], despite primary enamel is softer and less elastic than permanent enamel [27]. Our way of measuring caries experience aims to bridge the gap between caries experience in the primary and permanent dentition, as the ability to predict this information is limited [26]. The use of genetic variation as a surrogate of information on an individual’s enamel in primary dentition and therefore future caries experience in those teeth and possibly permanent dentition can influence a healthcare provider to choose a personalized prophylactic treatment plan for high-risk individuals. We predict that genomic approaches such as the one we are proposing coupled with the use of magnification systems for routine conservative operative dentistry [28] will define best practices of future dental professionals.

## Figures and Tables

**Figure 1 jpm-10-00005-f001:**
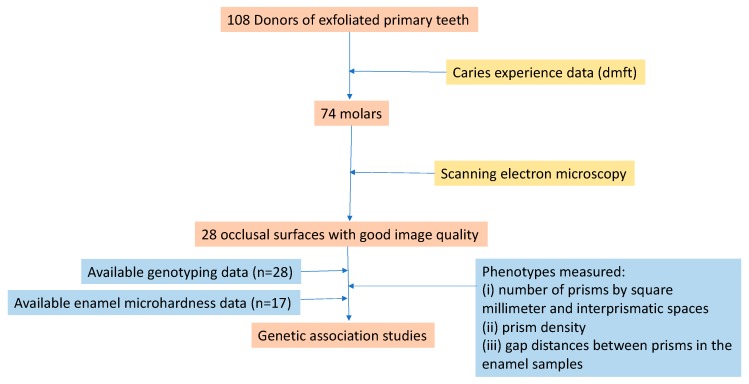
Summary of the study design.

**Figure 2 jpm-10-00005-f002:**
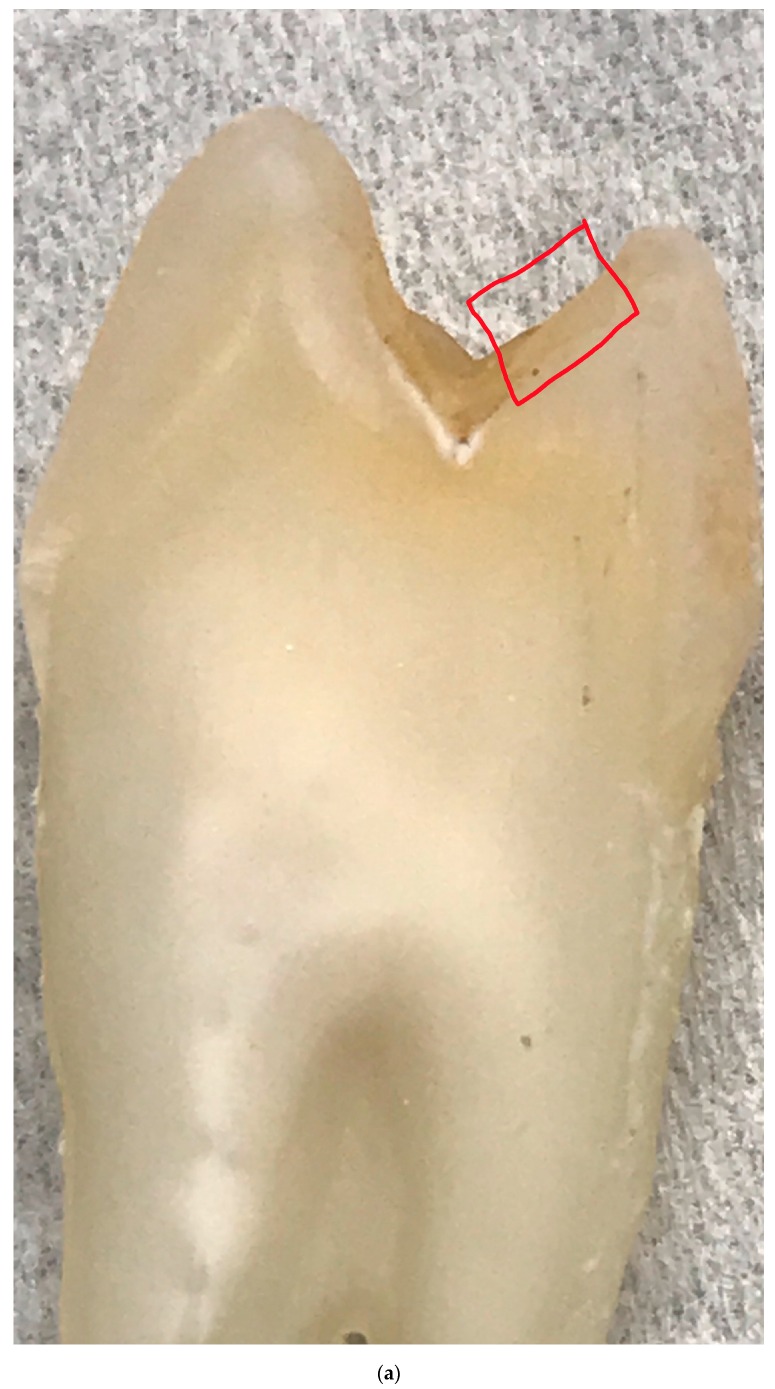
Teeth were sectioned mesio-distally, and a portion of the occlusal surface (**a**, marked in red) was selected so that the polished surface of the specimen (**b**, magnified image included in epoxy resin) parallel to the external surface was exposed for testing. After the test areas were selected and isolated, samples were polished, and then embedded in epoxy resin. We used a water-cooled coarse diamond saw to cut the specimen blocks. The plastic was ground away to the desired plane of section using a rotating wheel covered with an abrasive paper disc that was bathed in water. Each plastic block containing a prepared specimen was glued to the scanning electron microscope stub. Specimens were left to dry overnight and were coated with gold.

**Figure 3 jpm-10-00005-f003:**
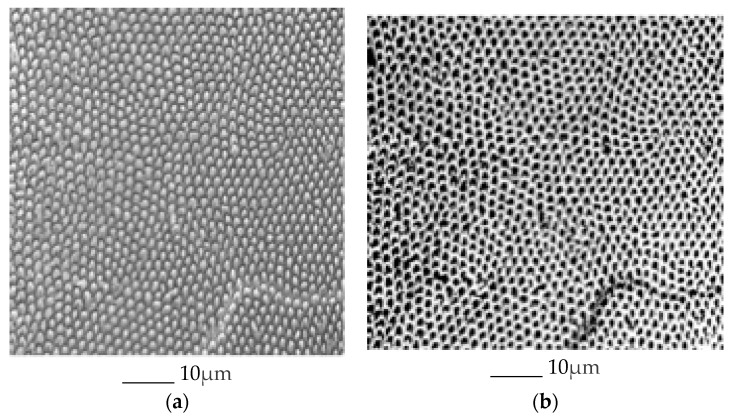
SEM micrographs of the prism/interprism continua before (**a**) and after (**b**) using the binary feature. The prisms and interprismatic spaces in the right image were counted using the analyze particles tool.

**Figure 4 jpm-10-00005-f004:**
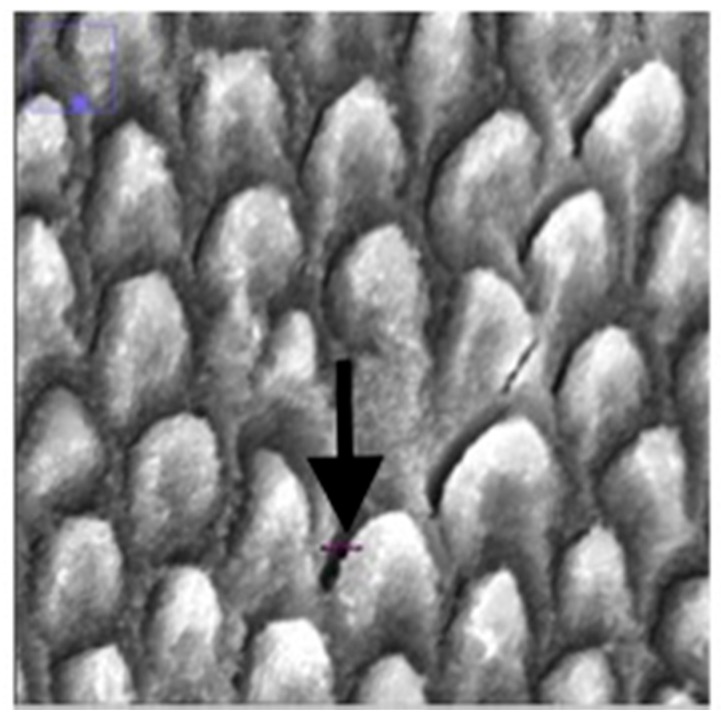
Enlarged SEM micrograph of enamel specimen revealing how gap distance was measured using the line feature on *ImageJ* (indicated by the black arrow).

**Figure 5 jpm-10-00005-f005:**
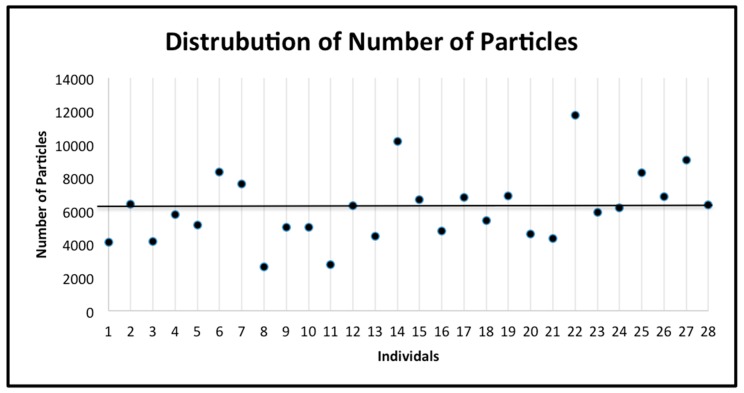
The distribution of the number of prisms by square millimeter for all individuals. The mean (6142.96) is indicated by the horizontal line. The standard deviation is 2071.24.

**Figure 6 jpm-10-00005-f006:**
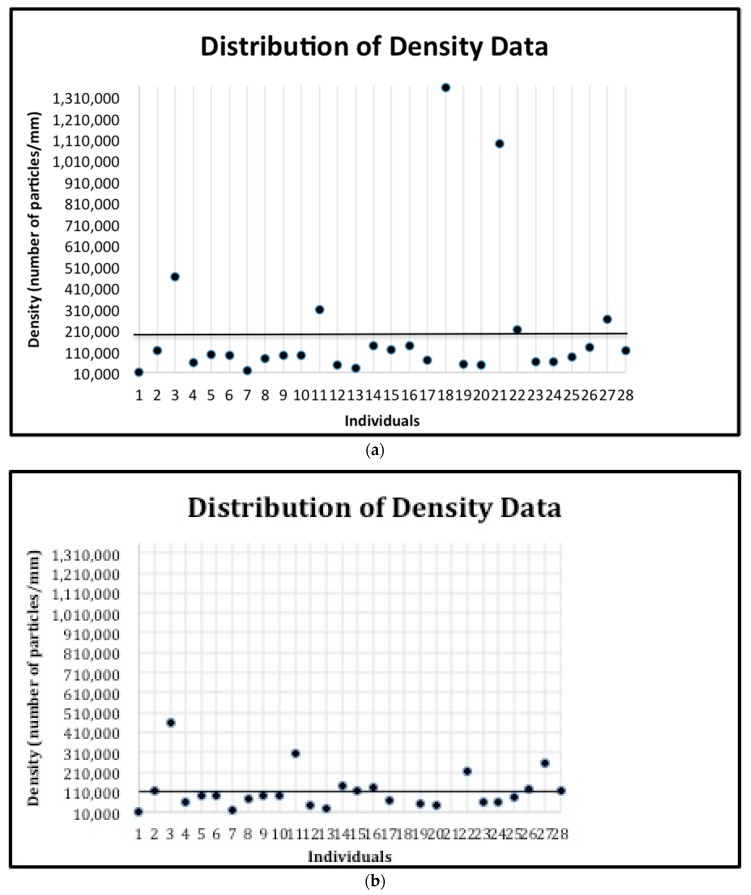
(**a**) The distribution of the prism density data for all individuals. The mean (192,791.89) is indicated by the horizontal line. The standard deviation is 307574.88. (**b**) The distribution of the prism density data for all individuals after removing the two outliers. The mean (113,766) is indicated by the horizontal line. The standard deviation is 99039.27.

**Figure 7 jpm-10-00005-f007:**
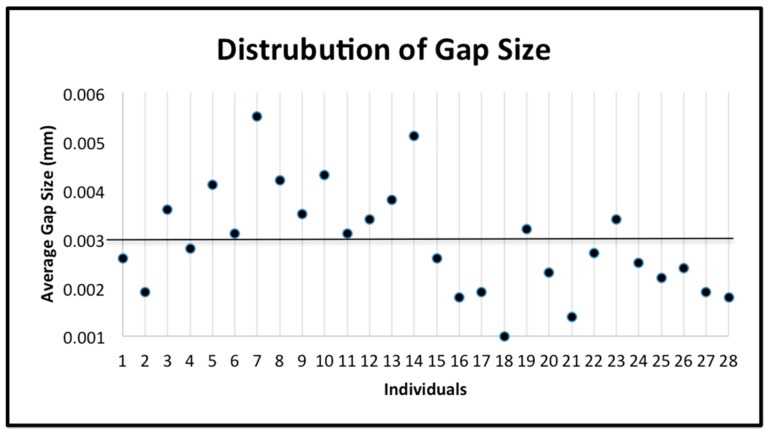
The distribution of the gap size data for all individuals. The mean (0.002932) is indicated by the horizontal line. The standard deviation is 0.001085.

**Table 1 jpm-10-00005-t001:** Characteristics of the population from whom the samples were obtained.

Characteristic	N (28 Donors of Primary Molar Occlusal Surfaces)
Mean age in years (standard deviation)	10.46 (4.65)
Sex	
Male	15
Female	13
Caries status of the individuals studied Mean dmft (standard deviation)	2.82 (2.67)
Enamel Microhardness (knoop) * Baseline (mean and standard deviation) After artificial caries creation (mean and standard deviation) After fluoride exposure (mean and standard deviation)	220.21 (55.07) 149.86 (58.97) 156.66 (63.80)

Values are given as mean (standard deviation) or sample size, dmft, decayed, missing, or filled teeth index for permanent dentition; dmft decayed, missing, or filled teeth index for primary dentition. * Enamel microhardness for the occlusal surface was only available for 17 samples. Differences in enamel microhardness in the three experimental conditions (at baseline, after artificial caries creation, and after fluoride exposure) were statistically significant (*p* < 0.001).

**Table 2 jpm-10-00005-t002:** Markers studied and the closest relative gene.

SNP	Closest Gene Symbol
rs2748729	-
rs4829728	-
rs4830231	*FIRRE*
rs5907830	-
rs5908778	-
rs5930702	-
rs5977872	*GPC4*
rs6637822	-
rs6638230	*PHF6*
rs6574293; rs10132091; rs745011; rs1077430; rs4903399	*ESRRB*
rs4903419	-
rs2860216	-
rs1676303	-
rs1997532	-
rs7150049	-
rs1997533	-
rs8011979	-
rs27565	*PART1*
rs4800418	-
rs6862039	*BTF3*
rs2287798	*AQP8*
rs1996315	*AQP6*
rs2878771	*AQP2*
rs461872	*AQP2*
rs2741559	*PIGT*
rs467323	*AQP2*
rs17159702	*AQP1*
rs4694075, rs34538475	*AMBN*
rs17878486; rs946252	*AMELX; ARHGAP6*
rs12640848; rs3796704	*ENAM*
rs7526319; rs3828054; rs2337360; rs3790506	*TUFT1*
rs5997096; rs134136	*TFIP11*
rs4970957	*MIR554; TUFT1*
rs11362	*DEFB1*
rs2619112; rs7217186	*ALOX15*
rs2235091; rs198968	*KLK4*
rs12156770	*SLITRK4*
rs1982	*PLAC1*
rs2097778	-
rs875459	*CCNB1*
rs1784418	*MMP20*
rs10875989	*AQP2*
rs3741559	*AQP2*
rs9466252	*CASC15*
rs4700418	*ZSWIM6*
rs3736309; rs296763	*AQP5*

*FIRRE*: functional intergenic repeating RNA element; *GPC4*: glypican 4; *PHF6*: PHD finger protein 6; *ESRRB*: estrogen related receptor beta; *PART1*: prostate androgen-regulated transcript 1; *BTF* 3: basic transcription factor 3; *AQP 1,2,5,6,8*: aquaporins 1,2,5,6,8; *AMBN*: ameloblastin; *AMLEX*: amelogenin; *ARHGAP6*: Rho GTPase activating protein 6; *ENAM*: emanelin; *TUFT1*: tuftelin; *TFIP11*: tuftelin-interacting protein 11; *MIR554*: microRNA 554; *DEFB1*: beta defensin 1; *ALOX15*: arachidonate 15-lipoxygenase; *KLK4*: kallikrein-related peptidase 4; *SLITRK4*: SLIT and NTRK like family member 4; *PLAC1*: placenta specific 1; *CCNB1*: cyclin B1; *MMP20*: matrix metalloproteinase 20; *CASC15*: cancer susceptibility 15; ZSWIM6: zinc finger SWIM-type containing 6; PIGT: phosphatidylinositol glycan anchor biosynthesis class T.

**Table 3 jpm-10-00005-t003:** Summary of single nucleotide polymorphisms and significant associations between caries experience and each phenotype for all data.

Marker	A1	F_A	F_U	A2	*p*-Value	Data Category and Phenotype
rs461872	A	0.5	0.2083	G	0.037	All P1
rs10132091	C	0.42	0.1786	T	0.049	
rs5930702	C	0.25	0.5714	G	0.05	
rs5908778	T	0.28	0.6842	C	0.01	
rs875459	T	0.8	0.4091	G	0.02	All P2 including outliers
rs3736309	C	0.2	0.55	G	0.05	
rs8011979	T	0	0.3889	C	0.03	
rs4800418	C	0.2	0.55	G	0.05	
rs6638230	A	0.57	0.1852	G	0.04	
rs3790506	G	0.33	0.04545	A	0.02	
rs467323	T	0.61	0.2308	C	0.01	
rs1997532	T	0.5	0.1667	C	0.01	All P2 excluding outliers
rs1997533	G	0.5	0.2188	C	0.04	
rs5977872	G	0.25	0.5789	A	0.05	
rs11362	T	0.5	0.2143	C	0.03	All P3
rs1997532	T	0.17	0.4286	C	0.04	
rs1997533	G	0.19	0.5	C	0.02	
rs2619112	G	0.67	0.3333	A	0.01	
rs7217186	T	0.56	0.1667	C	0.01	
rs4694075	T	0.83	0.25	C	0.03	1ST MU P1
rs5908778	T	0.5	0	C	0.04	
rs34538475	T	1	0.2	G	0.03	1ST MU P2
rs2287798	C	1	0.08333	G	0.003	
rs6637822	G	1	0.25	C	0.05	
rs4830231	T	1	0.2222	C	0.04	
rs6638230	A	0.5	0	G	0.03	
rs2097778	A	0.5	0	G	0.03	
rs12640848	G	0.75	0.1	A	0.01	1ST MU P3
rs17159702	T	1	0.125	C	0.004	
rs12156770	T	1	0.25	C	0.03	
rs5908778	T	0.67	0	C	0.01	
rs27565	C	0.62	0.2	T	0.03	2ND MU P1
rs9466252	T	0.5	0.15	C	0.05	
rs1784418	T	0.67	0.25	C	0.05	
rs1997532	T	0.62	0.2	C	0.03	
rs5977872	G	0	0.47	A	0.04	
rs3790506	G	0.5	0.07	A	0.04	2ND MU P2
rs11362	T	1	0.23	C	0.003	
rs7150049	G	0	0.57	A	0.04	
rs5930702	C	1	0.29	G	0.02	
rs4830231	C	0.22	0.7	T	0.04	2ND MU P3
rs5977872	G	0.54	0.1	A	0.03	
rs461872	A	0.5	0	G	0.03	1ST ML P1
rs5907830	G	1	0.25	C	0.05	
rs4903399	T	0.5	0	C	0.05	1ST ML P2
rs6637822	G	0	0.83	C	0.01	
rs5907830	G	0	0.67	C	0.03	
		NONE				1ST ML P3
rs11362	T	0.62	0.17	C	0.03	2ND ML P1
rs7150049	G	0.12	0.58	A	0.04	
rs10132091	C	0.67	0.17	T	0.03	
rs4829728	T	0.67	0.14	A	0.05	
rs5908778	T	0.17	0.71	C	0.05	
rs4830231	T	0.75	0.19	C	0.03	2ND ML P2
rs5997096	T	1	0.37	C	0.02	
rs1784418	T	0.2	0.7	C	0.02	2ND ML P3
rs5977872	A	0.14	0.8	G	0.02	
rs6638230	A	0.67	0	G	0.03	
rs2097778	A	0.57	0	G	0.04	
rs5907830	G	0.14	0.83	C	0.01	

A1: minor frequency allele; F_A: frequency of affected; F_U: frequency of unaffected.

**Table 4 jpm-10-00005-t004:** Summary of the significant correlations (r = 0.4) between caries experience and each phenotype, microhardness and each phenotype, and between the three phenotypes.

	Phenotype 1 (r)	Phenotype 2 (r)	Phenotype 3 (r)
2nd MU			
LCE	0.11	−0.38	0.24
HCE	−0.71	NC	NC
All Data			
Baseline	NC	−0.63	NC
Lesion	NC	NC	NC
Fluoride	NC	NC	NC
2nd MU			
Baseline	N/A	N/A	N/A
Lesion	N/A	N/A	−0.7
Fluoride	N/A	N/A	−0.81
All Data			
Phenotype 1	NC	NC	−0.45
1st MU			
Phenotype 2	NC	NC	−0.45
2nd MU			
Phenotype 2	NC	NC	−0.46

LCE: low caries experience; HCE: high caries experience; 1ST MU: first upper molar; 1ST ML: first lower molar; 2ND MU: second upper molar; 2ND ML: second lower molar; NC: no significant correlation; N/A: not enough data to perform correlation. The number of prisms by square millimeter was classified as phenotype 1. The prism density was classified as phenotype 2. The gap size was classified as phenotype 3.

**Table 5 jpm-10-00005-t005:** Breakdown of individuals with the minor allele and statistically significant associations to our phenotypes that match our hypothesis regarding number of prisms by square millimeter, prism density, and gap distance and caries experience.

Number of SNPs Total	87
Number of SNPS with no results matching our hypothesis	5
Number of SNPs with matching results	82
Number of SNPs with 1–4 individuals	49
Number of SNPs with 5–8 individuals	28
Number of SNPs with 9–12 individuals	5
Percentage of individuals that match the hypothesis	94.25

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
