# Peer review of "Measuring the Microscopic Structures of Human Dental Enamel Can Predict Caries Experience"

_jpm, 2020, doi:10.3390/jpm10010005_

Round 1

Reviewer 1 Report

I really think that this paper has several merits.

I have just a small comment 

change the term decay with caries

Author Response

Thanks for carefully reviewing our work. We edited the text and substituted the term "decay" for "caries" where pertinent. We marked those changes in the resubmitted text in yellow.

Reviewer 2 Report

Overall comment

Thank you for the opportunity to review this manuscript, which aims to investigate genetic risk factors for dental caries by using phenotypes based on enamel characteristics. I can certainly see the value and novelty of the approach applied in this study which aims to better understand the mechanisms that may underlie a genetic pre-disposition to dental caries and overcome some of the problems regarding phenotype definition in genetic investigations into dental caries. However, the manuscript in its current form needs some restructuring or clarification to be readily understood. The methodology is quite complex, involving clinical, genetic and laboratory measurements and analysis of correlations between these and the main hypothesis and the methodology applied is unclear. I have provided some brief specific points relating to the introduction and methods. However, the main issue, which I suggest can be overcome with better reporting, is lack of clarity about the research question, and analysis undertaken to answer the question. Specifically, the different samples sizes (from 108 to 28 to 17) are difficult to follow – I have suggested a flow chart demonstrating which analyses involved the entire cohort of 108 participants would be useful as it seems that most of the analysis is based on microhardness and the values derived from SEM images, both performed on a small subset. The key phenotypes are only stated towards the latter parts of the methods but should be stated (even briefly) in the aims or at least at the start of the methods, prior to detail about genetic analyses etc. The lack of clarity about the methods makes it difficult to fully appraise the results and particularly understand the limitations of the study. However, the discussion seems appropriate but should also note the issue about DDE which does not appear to have been measured in this study.

Abstract

Page 1, Line 30 -32 “Scanning electron microscopy (SEM) images of enamel from primary teeth  were used to measure number of prisms by square millimeter and interprismatic spaces, prism density and gap distances between prisms in the enamel samples”

Please rephrase/annotate this sentence as it is difficult to follow – I suggest using numbers to specify the three outcomes/characteristics as done later in the main methods.

Introduction

Page 2, Line 95 – 96 “The design is inspired in the evidence that caries experience in the primary dentition can predict to a certain degree caries experience in the permanent dentition.”

This sentence should potentially be included earlier in the background but I am not sure of the relevance to the research question – it either needs to be explained further or removed.

Materials and Methods

Page 30

Enamel Specimens – Was there any consideration given to developmental defects of enamel and how these could influence the findings of the study? Was this measured as part of the examination – DDE could be an important confounder of the relationships between dental caries, genetics and enamel properties.

Page 3, Line 125 - 126 “Twenty-eight of the 74 molars that had a sound occlusal surface were selected and used for further microscopic analysis since they were considered to be sectioned at the same orientation.”

Microscopic analysis seems to be a key component of this study, but seems to have been performed on a much smaller subset than initially suggested with the 108 participants. A flow chart showing the number of participants in each component of the study and which analyses involved the entire 108 group would be useful.

Page 3, Line 127 “Characteristics of the population studied are presented in Table 1”

These are results and so would be more appropriate in the results section.

Page 3, Table 1: N =28 in this table but the methods suggest that microhardness was only measured in 17 samples – therefore it is more appropriate to list the characteristics of the samples population.

Page 4, Line 134 “*Enamel microhardness for the occlusal surface was only available for 17 samples.”

Again, this seems to suggest that enamel microhardness was only evaluated in 17 of the108 participants – please see my comment earlier and clarify each component of the analysis and the number of participants/samples in each.

Page 4, Line 134 -136 “Differences in enamel microhardness in the three experimental conditions (at baselines, after artificial caries creation, and after fluoride exposure) were statistically significant (P<0.05).

Please specify the actual p value, unless p<0.001. This sentence seems to be reporting results and therefore should be moved from the methods. Please also clarify if this relates to the mean microhardness of the entire sample – perhaps refer to Table 1 if relevant and include the p values in the table.

Page 5, Line 160 “Correlation tests with previously studied phenotypes”

Please state the actual phenotypes in the title rather than referring to a previous study.

This section starts with statistical analysis but comprises mostly details about laboratory specimen preparation – I suggest leaving the statistical analysis to a separate section later and changing the title to reflect the main message in this section.

Page 5, Line 162-163 “he subjects were classified as having “low caries experience” (below the mean DMFT+ dmft score of the 28 subjects) or “high caries experience” (above the mean DMFT+dmft score of the 28 subjects).

Why were these determined based on the 28 subjects – it seems more appropriate to either use a dmft value that reflects the population (ie from the larger sample of 108 participants) or of the actual study sample of 17.

 Page 5, Line 163-164 “each measurement phenotype explained above”

Please clarify the 'measurement phenotypes'. The section immediately prior relates to the SNPs and so this is not clear.

Author Response

Here, point-by-point, the answers for the concerns. Changes in the text are marked in yellow.

Abstract

Page 1, Line 30 -32 “Scanning electron microscopy (SEM) images of enamel from primary teeth  were used to measure number of prisms by square millimeter and interprismatic spaces, prism density and gap distances between prisms in the enamel samples”

Please rephrase/annotate this sentence as it is difficult to follow – I suggest using numbers to specify the three outcomes/characteristics as done later in the main methods.

RESPONSE: We made the suggested change.

Introduction

Page 2, Line 95 – 96 “The design is inspired in the evidence that caries experience in the primary dentition can predict to a certain degree caries experience in the permanent dentition.”

This sentence should potentially be included earlier in the background but I am not sure of the relevance to the research question – it either needs to be explained further or removed.

RESPONSE: We added a clarifying statement as suggested.

Materials and Methods

Page 30

Enamel Specimens – Was there any consideration given to developmental defects of enamel and how these could influence the findings of the study? Was this measured as part of the examination – DDE could be an important confounder of the relationships between dental caries, genetics and enamel properties.

RESPONSE: Samples did not have any DDE. We added a clarifying statement as suggested.

Page 3, Line 125 - 126 “Twenty-eight of the 74 molars that had a sound occlusal surface were selected and used for further microscopic analysis since they were considered to be sectioned at the same orientation.”

Microscopic analysis seems to be a key component of this study, but seems to have been performed on a much smaller subset than initially suggested with the 108 participants. A flow chart showing the number of participants in each component of the study and which analyses involved the entire 108 group would be useful.

RESPONSE: We added the flowchart as suggested.

Page 3, Line 127 “Characteristics of the population studied are presented in Table 1”

These are results and so would be more appropriate in the results section.

RESPONSE: Since we mentioned the demographic characteristics of the sample at this point in the text, we made reference of Table 1 already on page 3.

Page 3, Table 1: N =28 in this table but the methods suggest that microhardness was only measured in 17 samples – therefore it is more appropriate to list the characteristics of the samples population.

RESPONSE: We had added originally a note to clarify this discrepancy. Not all 28 had microhardness data.

Page 4, Line 134 “*Enamel microhardness for the occlusal surface was only available for 17 samples.”

Again, this seems to suggest that enamel microhardness was only evaluated in 17 of the108 participants – please see my comment earlier and clarify each component of the analysis and the number of participants/samples in each.

RESPONSE: WE added the suggested flowchart.

Page 4, Line 134 -136 “Differences in enamel microhardness in the three experimental conditions (at baselines, after artificial caries creation, and after fluoride exposure) were statistically significant (P<0.05).

Please specify the actual p value, unless p<0.001. This sentence seems to be reporting results and therefore should be moved from the methods. Please also clarify if this relates to the mean microhardness of the entire sample – perhaps refer to Table 1 if relevant and include the p values in the table.

RESPONSE: We edited the p-value as suggested. The table is showing the values used for the analyses.

Page 5, Line 160 “Correlation tests with previously studied phenotypes”

Please state the actual phenotypes in the title rather than referring to a previous study.

RESPONSE: We edited the title as suggested. 

This section starts with statistical analysis but comprises mostly details about laboratory specimen preparation – I suggest leaving the statistical analysis to a separate section later and changing the title to reflect the main message in this section.

RESPONSE: We originally described each step in separate and believe it was better to refer to the correlation in separate from the association tests done later.

Page 5, Line 162-163 “he subjects were classified as having “low caries experience” (below the mean DMFT+ dmft score of the 28 subjects) or “high caries experience” (above the mean DMFT+dmft score of the 28 subjects).

Why were these determined based on the 28 subjects – it seems more appropriate to either use a dmft value that reflects the population (ie from the larger sample of 108 participants) or of the actual study sample of 17.

RESPONSE: All analyses obtained from the images were performed in the 28 individuals, with the only exception when microhardness data was looked at, which was available only for 17, but most calculations were performed with the 28.

 Page 5, Line 163-164 “each measurement phenotype explained above”

Please clarify the 'measurement phenotypes'. The section immediately prior relates to the SNPs and so this is not clear.

RESPONSE: We revised the sentence as requested.

This manuscript is a resubmission of an earlier submission. The following is a list of the peer review reports and author responses from that submission.

Round 1

Reviewer 1 Report

I appreciate the authors' attempt to present support for an alternative etiologic explanation for caries, but am concerned about the underlying data and approach. Overall, I suggest a large re-organization that includes (a) articulating the role of genetics in the conceptual framework and relate this to the analytic approach chosen, (b) consideration of a more robust statistical approach that leverages more of the available data, (c) revised presentation of results, and (d) overall clarification of the Methods employed.

A major overarching concern I have about this analysis is that primary tooth enamel is being evaluated in relation to permanent tooth caries outcomes. Given established literature around the influence of environmental factors on the developing enamel, investigation of this relationship is inappropriate unless framed and articulated exactly for what it is (with limitations clearly stated).

Below please find suggestions for specific improvements.

Abstract: 

list Results in a consistent order throughout the manuscript, in relation to exposures and outcomes (eg, caries, hardness (baseline, lesion, fluoride)) Specific findings (Results) should better support the "Conclusion" as stated  the Clinical Relevance is much appreciated

Introduction

articulate better how the genetic component of this analysis is evaluated - the current focus of the manuscript is on the structure-decay relationship conceptual framing (with a figure) of factors evaluated would be helpful - it is not made clear how the genetic measures are being considered in relation to enamel structure and DMFT 

Methods

89=90: introduce the study population/ sample 81-87: molars only were evaluated or molars, incisors, and canines? 89-90: were any environmental factors considered? There is a lot of individual variation in environmental exposures in subjects from a single location 93-95: How was the study sample selected? Was there exclusion/inclusion criteria? How were teeth collected? 93-95: Was an oral exam done? 100: What treatments were performed in sound teeth? Was age and number of permanent teeth adjusted for in analyses looking at DMFT as an outcome?  All tables and data should be presented in the Results section (not Methods or Discussion) use of dfmt v DMFT should be clear and consistent throughout the manuscript / more clearly articulate how the dmft v DMFT was dealt with 138-139: better describe the selection process for the sample of teeth evaluated 149-156: excellent description of the use of the ImageJ software! Fig 1: higher quality images should be provided Figures 4-6: should be in the Results section more clearly articulate the difference between phenotypes 1 and 2 241-243: how and why were blocks in root surfaces chosen as controls? 238- : tooth sample prep should be higher up in the Methods 245-247: citation for this method of mimicking the demineralization process? 248-267: what was the timing of the artificial de/remineralization steps? Did the teeth all remineralize to the same degree? include justification for creating phenotype groups and how thresholds were chosen - why not evaluate continuous enamel measures? are there 3 or 4 phenotypes (seems inconsistent)? better articulate the difference between prism density and number of prisms per square millimeter

Results

278: "associations" were not evaluated (only "correlations") Why were no multivariate analyses done? Correlations alone seem an inappropriate approach to this topic please further discuss the evaluation of the gene markers check if Table 3 is labeled correctly Table 4 - why is most of the table NA? Why not show all the teeth together given small sample size? a concern is the assumption inherent in the analysis that one tooth represents whole mouth (enamel structure of one tooth is evaluated in relation to DMFT) and the hypothesis that primary tooth enamel predicts DMFT is, at best, not a sensitive analysis and, at worst, flawed - this should be addressed  Another concern is that the study lacks power to detect minor associations 

Author Response

I appreciate the authors' attempt to present support for an alternative etiologic explanation for caries, but am concerned about the underlying data and approach. Overall, I suggest a large re-organization that includes (a) articulating the role of genetics in the conceptual framework and relate this to the analytic approach chosen, (b) consideration of a more robust statistical approach that leverages more of the available data, (c) revised presentation of results, and (d) overall clarification of the Methods employed.

RESPONSE: We addressed these concerns below.

A major overarching concern I have about this analysis is that primary tooth enamel is being evaluated in relation to permanent tooth caries outcomes. Given established literature around the influence of environmental factors on the developing enamel, investigation of this relationship is inappropriate unless framed and articulated exactly for what it is (with limitations clearly stated).

RESPONSE: We added rationale for designing our study the way the reviewer described both in the introduction and discussion sections.

Abstract: 

list Results in a consistent order throughout the manuscript, in relation to exposures and outcomes (eg, caries, hardness (baseline, lesion, fluoride)) Specific findings (Results) should better support the "Conclusion" as stated  the Clinical Relevance is much appreciated

RESPONSE: There are many associations and correlations and the abstract was written more generally to accommodate the limited space we have.

Introduction

articulate better how the genetic component of this analysis is evaluated - the current focus of the manuscript is on the structure-decay relationship conceptual framing (with a figure) of factors evaluated would be helpful - it is not made clear how the genetic measures are being considered in relation to enamel structure and DMFT 

RESPONSE: We added additional commentary in the introduction to address this concern.

Methods

89=90: introduce the study population/ sample

RESPONSE: This was originally done.

81-87: molars only were evaluated or molars, incisors, and canines?

RESPONSE: Only SEM images of occlusal surfaces of molars were used for analyses.

89-90: were any environmental factors considered? There is a lot of individual variation in environmental exposures in subjects from a single location

RESPONSE: We made the assumption there were not many differences that could dramatically affect the phenotypes defined in the study.

93-95: How was the study sample selected? Was there exclusion/inclusion criteria? How were teeth collected?

RESPONSE: All families invited to participate that donated teeth were included. No pre-selection was made.

93-95: Was an oral exam done?

RESPONSE: Yes. The information was available for this study.

100: What treatments were performed in sound teeth?

RESPONSE: Typically professional polishing and fluoridated gel application.

Was age and number of permanent teeth adjusted for in analyses looking at DMFT as an outcome? 

RESPONSE: Not in this limited sample.

All tables and data should be presented in the Results section (not Methods or Discussion) use of dfmt v DMFT should be clear and consistent throughout the manuscript / more clearly articulate how the dmft v DMFT was dealt with

RESPONSE: We find appropriate to refer to a table when it feels necessary in the text. As we have done in previous publications, we introduce tables in the methods or discussion sections. We originally described how dmft and DMFT or dealt with around line 405 in the discussion session.

138-139: better describe the selection process for the sample of teeth evaluated

RESPONSE: We included all tooth images we could use as originally described.

149-156: excellent description of the use of the ImageJ software! Fig 1: higher quality images should be provided Figures 4-6: should be in the Results section more clearly articulate the difference between phenotypes 1

RESPONSE: We repeated the definition of  the phenotypes as suggested. Unfortunately, figures are presented in their original format.

and 2 241-243: how and why were blocks in root surfaces chosen as controls?

RESPONSE; This was a description we meant to originally delete. It was deleted.

238- : tooth sample prep should be higher up in the Methods

RESPONSE: We listed last the data already existed to avoid confusion to the experiments done afterwords, which are reported here. We changed the order as requested.

245-247: citation for this method of mimicking the demineralization process?

RESPONSE. We originally included a citation (reference 2).

248-267: what was the timing of the artificial de/remineralization steps? Did the teeth all remineralize to the same degree? include justification for creating phenotype groups and how thresholds were chosen - why not evaluate continuous enamel measures? are there 3 or 4 phenotypes (seems inconsistent)? better articulate the difference between prism density and number of prisms per square millimeter

RESPONSE; Artificial caries was created as soon as all samples were available for study. As in previous work, we used the mean value of the sample to define the two comparison groups. We focused on 3 phenotypes, not four.

Results

278: "associations" were not evaluated (only "correlations") Why were no multivariate analyses done?

RESPONSE:When we compare the distribution of genotypes or alleles and this distribution is different, we used the term association, since a particular allele frequency associate with a particular status. correlations we used to suggest if obtained values (higher or lower) correlated to other outcomes (as being high or low).

Correlations alone seem an inappropriate approach to this topic please further discuss the evaluation of the gene markers check if Table 3 is labeled correctly

REPONSE: We only used correlations for instances other than genotyping data. Table 3 is labelled correctly.

Table 4 - why is most of the table NA?

RESPONSE: Many comparisons are not statistically different.

Why not show all the teeth together given small sample size?

RESPONSE: We just included an evaluation the occlusal surface of a posterior tooth per individual..

a concern is the assumption inherent in the analysis that one tooth represents whole mouth (enamel structure of one tooth is evaluated in relation to DMFT) and the hypothesis that primary tooth enamel predicts DMFT is, at best, not a sensitive analysis and, at worst, flawed - this should be addressed  Another concern is that the study lacks power to detect minor 

RESPONSE: We added discussion as suggested to address this concern.

Reviewer 2 Report

I really think that this paper merits to be accepted maybe after some minor revisions.

The weak points of the study in my opinion are:

No mention about fluoride in the water or the concentration of fluoride into the tooth structure;

How to put the results of this study in the clinical practice and in the public health field.

I am not sure that the use of dmft was the correct choice to evaluate caries experience in a study like that; probably I would have preferred the comparison between ICDAS or even better with a diagnostic/detection method allowing to discriminate between active and no-active lesions

Author Response

No mention about fluoride in the water or the concentration of fluoride into the tooth structure;

RESPONSE: We did not have these data. We added a comment in the methods section for clarity.

How to put the results of this study in the clinical practice and in the public health field.

RESPONSE: We made an attempt originally in the end of the manuscript to describe that.

I am not sure that the use of dmft was the correct choice to evaluate caries experience in a study like that; probably I would have preferred the comparison between ICDAS or even better with a diagnostic/detection method allowing to discriminate between active and no-active lesions

RESPONSE; ICDAS was not available for this historical samples. ICDAS still carries the limitation of describing active lesions only, and may not provide a better picture of past caries experience.

Round 2

Reviewer 1 Report

inadequate consideration of original review